# Large, Nested Variant of Urothelial Carcinoma Is Enriched with Activating Mutations in Fibroblast Growth Factor Receptor-3 among Other Targetable Mutations

**DOI:** 10.3390/cancers15123167

**Published:** 2023-06-13

**Authors:** Yaser Gamallat, Mitra Afsharpad, Soufiane El Hallani, Christopher A. Maher, Nimira Alimohamed, Eric Hyndman, Tarek A. Bismar

**Affiliations:** 1Department of Pathology and Laboratory Medicine, Cumming School of Medicine, University of Calgary, Calgary, AB T2N 4N1, Canada; 2Department of Oncology, Biochemistry and Molecular Biology, Arnie Charbonneau Cancer Institute, Cumming School of Medicine, University of Calgary, Calgary, AB T2N 4N1, Canada; 3Alberta Precision Laboratory, Department of Laboratory Medicine and Pathology, Faculty of Medicine and Dentistry, University of Alberta, Edmonton, AB T6G 2R7, Canada; 4Division of Oncology, Department of Medicine, Siteman Cancer Center, Washington University School of Medicine, St. Louis, MO 63110, USA; 5Department of Oncology, Tom Baker Cancer Centre, University of Calgary, Calgary, AB T2N 4N2, Canada; 6Department of Surgery and Urology, Cumming School of Medicine, University of Calgary, Calgary, AB T2N 2T9, Canada; 7Alberta Precision Laboratory, Rockyview General Hospital, Departments of Pathology and Laboratory Medicine, Calgary, AB T2V 1P9, Canada

**Keywords:** large nested variant of urothelial carcinoma (LNVUC), *fibroblast growth factor receptor-3* (*FGFR-3*), metastatic urothelial carcinoma, whole-exome sequencing, targeted sequencing

## Abstract

**Simple Summary:**

Urothelial carcinoma of the large, nested variant is a specific histological morphology subtype of urothelial carcinoma. Although it is a rare variant, it requires specific attention due to its bland histology and the fact that it may potentially be missed in routine biopsies. In this study, we identify *Fibroblast Growth Factor Receptor-3 *(*FGFR-3*) as the most common mutation present in this subtype among other potential targetable mutations. All our cases of this variant also harbored other potentially actionable mutations in other genes, which could also be amenable to novel targeted therapy. Patients with this variant may benefit from additional molecular screening to identify potential therapeutic targets that could improve the clinical outcome of such patients.

**Abstract:**

The large, nested variant of urothelial carcinoma (LNVUC) is characterized by bland histomorphology mimicking that of benign von Brunn nests. In the current study, we aimed to investigate the *Fibroblast Growth Factor Receptor-3 (FGFR-3*) activation and missense mutation in 38 cases, including 6 cases diagnosed with LNVUC and 32 with metastatic invasive urothelial carcinoma (UC). Initially, six formalin-fixed paraffin-embedded (FFPE) tissue samples of the LNVUC were subjected to whole-exome sequencing (WES), and then we performed targeted sequencing on 32 cases of metastatic invasive UC of various morphological subtypes, which were interrogated for the *FGFR3*. Our results revealed 3/6 (50%) LNVUC cases evaluated by WES in our study showed an activating mutation in *FGFR-3*, 33% showed an activating mutation in *PIK3CA*, and 17% showed activating mutation in *GNAS* or *MRE11*. Additionally, 33% of cases showed a truncating mutation in *CDKN1B*. All LNVUC in our study that harbored the *FGFR-3* mutation showed additional activating or truncating mutations in other genes. Overall, 6/32 (18.75%) cases of random metastatic invasive UC showed missense mutations of the *FGFR-3* gene. The LNVUC variant showed the higher incidence of *FGFR-3* mutations compared to other types of mutations. Additionally, all LNVUC cases show additional activating or truncating mutations in other genes, thus being amenable to novel targeted therapy.

## 1. Introduction

Urothelial carcinoma is the most common pathological (histological) type of bladder cancer [1]. Approximately 75% of cases are classified as pure urothelial carcinoma and 25% as variants of mixed or less common variants such as squamous and adenocarcinoma [2]. The recent 2022 edition of WHO recognized several of these urothelial carcinoma variants as having important prognostic and predictive value [3]. 

The nested variant of urothelial carcinoma is a very rare subtype that is characterized by bland nests of neoplastic urothelial cells that are often seen invading muscularis propria [2]. The large, nested variant (LNVUC) is defined as a spectrum of the nested variant and is characterized by larger nests of invasive neoplastic urothelial cells [4]. The nested variant of urothelial carcinoma, and the large, nested variant as a spectrum, was first described in 1979 but was not recognized by the WHO until 2004. The bland histomorphology of this variant can mimic those of a benign von Brunn nests, leading to misdiagnosis, particularly in small biopsies, causing a delay in recognizing this lesion and therefore the treatment of patients [5]. Studies have shown that despite bland histomorphology, this variant is associated with poor outcome [6,7]. Moreover, at the time of diagnosis, approximately 69% of nested variants are in advanced stages (pT3/4), and 19% have nodal involvement. As with all variants subtypes of urothelial carcinoma, the risk of recurrence and progression of the nested variant is increased based on many factors including the presence of a residual tumor and in situ component [8], which, understandably, can be easily missed in this variant due to the bland morphological appearance. 

*Fibroblast growth factor receptor-3 (FGFR-3)* is one of the four highly conserved fibroblast growth factor receptor tyrosine kinases (RTKs) and has been shown to have a role in tumor growth and survival regulations [9,10,11]. *FGFR-3* has been implicated in multiple malignant neoplasms including multiple myeloma, urothelial and cervical carcinoma [12,13,14]. Despite this, several studies have focused on *FGFR-3* as therapeutic and prognostic marker in urothelial carcinoma [10,15]. The clinical significance of *FGFR* mutations is highlighted with the development of the pan-*FGFR* inhibitor, Erdafitinib, and Pembrolizumab, which was approved by the Food and Drug Administration [13,16,17]. This has provided new hope in the management and treatment of malignancies with *FGFR-3* mutations [18].

Studies have shown *FGFR-3* mutation to be among the most common mutated oncogene in urothelial carcinoma overall [19]. *FGFR-3* mutations have been reported in about 75% of noninvasive urothelial carcinoma and about 15–20% of high-grade tumors [20]. However, most studies have focused on the presence of this mutation in the plasmacytoid variant of urothelial carcinoma, another rare and aggressive variant of urothelial carcinoma [21,22,23], but only a few have focused on the nested variant. 

This study is designed to characterize the molecular backgrounds of the large, nested variant of urothelial carcinoma, in the hope of identifying potential targetable mutations linking this specific histomorphology with specific genetic mutations.

## 2. Material and Methods

### 2.1. Patient Samples

In the current study, we recruited 38 metastatic UC samples of which 6 cases were diagnosed with LNVUC at Alberta Precision labs/University of Calgary Cumming School of Medicine between 2015 and 2019. Of these 6 cases, 2 belonged to the same patient (cases 2 and 3), who developed a primary LNVUC in the lower urothelial system (bladder) which was treated with partial cystectomy, and subsequently, after this diagnosis, he developed another primary LNVUC in the upper urothelial system (ureter) which was treated with bilateral nephroureterectomy. These cases were re-reviewed by an experienced genitourinary pathologist to confirm the diagnosis of this variant and select the most suitable areas for sequencing. The remaining 32 cases of metastatic invasive urothelial carcinoma, including 3 cases of the LNVUC, were also analyzed for *FGFR-3* target seq.

All cases were classified based on the 2016 edition of the World Health Organization Classification of Tumors of the Urinary System. LNVUC cases demographic data, stage at the time of diagnosis, treatments and outcomes up to date were documented for each case (Table 1), and further demographics about the 32 cases of metastatic invasive urothelial carcinoma cases are provided in Table 2.

### 2.2. DNA Extraction and Whole-Exome Sequencing

Pathologically reviewed samples confirmed the diagnosis and determined tumor content, and marked tumor areas on hematoxylin and eosin (H&E) slides used to accurately obtain tissue samples from formalin-fixed paraffin-embedded (FFPE) samples. Briefly, the study pathologists cored or scrolled the selected tumor areas. Then tumor DNA was extracted using the QIAamp DNA FFPE Tissue kit (Qiagen, Cat # 56404, Hilden, Germany). Germline DNA was extracted from normal kidney tissue adjacent to the tumor. Qubit used to quantify the DNA using Qubit DNA HS assay (Life Technologies, Carlsbad, CA, USA). Whole-exome sequencing (WES) was carried out by The Centre for Applied Genomics (TCAG), Toronto, ON, Canada. Briefly, 750 ng of DNA was used for WES exome library preparation and sequencing using SureSelect XT Human All Exon V5 Kit (Agilent Technologies, Santa Clara, CA, USA). Genomic DNA was fragmented to 200-bp on average using a Covaris LE220 instrument. Prior to ligation, sheared DNA was end-repaired, and the 3′ ends were adenylated on adapters with overhang-T. Then, the genomic library was amplified by PCR using 10 cycles and hybridized with biotinylated probes that target exonic regions; the enriched exome libraries were amplified by an additional 8 cycles of PCR. The exomic libraries were validated using DNA High-Sensitivity chips on a Bioanalyzer 2100 (Agilent Technologies) for size and by qPCR using the Kapa Library Quantification Illumina/ABI Prism Kit protocol (KAPA Biosystems) for quantities. Exome libraries were pooled and sequenced with the TruSeq SBS sequencing chemistry using a V4 high throughput flowcell on a HiSeq 2500 platform (Illumina Inc., San Diego, CA, USA), as per Illumina’s recommended protocol.

### 2.3. Data Alignment and Validation

Around 6–8 gigabases of raw paired end data of 126-bases were generated per exome library. Reads were aligned to the hg19 build human reference genome using BWA (version 0.5.9). PCR duplicates were marked using picard-tools-1.108, and local re-alignment and base recalibration were performed using GATK 1.1-28. Variants (SNV, indel) were called using GATK UnifiedGenotyper 1.1-28. An Annovar-based pipeline was used for adding gene-based, feature-based and frequency-based annotations for variant filtering and prioritization [24].

### 2.4. Targeted Sequencing for FGFR Gene Fusions

Targeted seq testing was performed at Cancer Genetics Clinic, Jewish General Hospital, using an NGS panel which analyzes both DNA and RNA extracted from FFPE material and detects sequence changes involving the following *FGFR* Fusion proteins: Driver genes partner genes ***FGFR1***: *ADAM32*, *BAG4*, *BCR*, *CNTRL*, *CPSF6*, *CUX1*, *ERVK3-1*, *FGFR1OP*, *FGFR1OP2*, *FN1*, *LRRFIP1*, *MYO18A*, *NTM*, *PLAG1*, *RANBP2*, *SQSTM1*, *TACC1*, *TPR*, *TRIM24*, *WHSC1L1*, *ZMYM2*, *ZNF703 **FGFR2***: *AFF3*, *AHCYL1*, *BICC1*, *CASP7*, *CCAR2*, *CCDC6*, *CD44*, *CIT*, *COL14A1*, *CREB5*, *CTNNB1*, *FAM76A*, *KCTD1*, *MGEA5*, *NOL4*, *OFD1*, *PARK2*, *PDHX*, *PPHLN1*, *SHTN1*, *SLC45A3*, *SNX19*, *TACC3*, *TXLNA*, *USP10 **FGFR3***: *AES*, *BAIAP2L1*, *ELAVL3*, *ETV6*, *FBXO28*, *JAKMIP1*, *TACC3.*

## 3. Results

Whole-exome sequencing results (Table 3) for the six cases of the large, nested variant of urothelial carcinoma showed 3 cases (50%) to harbor positive activating mutation in *FGFR-3*. Two cases showed an activating mutation in *PIK3CA* (33%), and one showed activating mutations in GNAS (17%), and another case showed mutations in *MRE11* (17%). 

A truncating mutation in CDKN1B was seen in two cases (33%), and truncating mutations in *CDKN2A*, *ARID1B*, *ARID1A* and *KDM6A* was seen only in one case each (17%). It was interesting to note that cases that showed activating mutations in *FGFR-3* also showed additional activating and truncating mutations in other genes including *PIK3CA*. Detailed genetic mutations detected in the six cases studied are presented in Table 3.

All our cases that harbored *FGFR-3* mutations showed additional activating or truncating mutations. One case showed simultaneous mutations in *FGFR-3*, *PIK3CA*, *CDKN1B*, *ARID1B* and *PPP2R1A*. Another case showed mutations in *FGFR-3* as well as *MRE11* and *KDM66*, and an additional case had simultaneous mutations in *FGFR-3*, *CDKN1B* and *CDKN2A*. Of the cases that did not harbor any *FGFR-3* mutations, one case showed simultaneous mutations in *PIK3CA*, *ARID1A* and *GNAS* (Table 3).

As exhibited in Table 2, *FGFR-3* mutation fusions by targeted sequencing were assessed in the 32 cases, including three cases of the six WES-performed samples series, which did not show *FGFR-3* mutations by WES. Of those, 6/32 cases (18%) were positive for *FGFR-3* missense mutations (two *S249C*, three *Y373C* and one *G370C* mutations). Three out of six cases exhibited dedifferentiated histology (poorly differentiated or squamous/sarcomatoid differentiation), five of six were from metastatic sites, and one was from the upper urothelial tract. However, no significant association was noted to specific histopathological morphology or metastatic site.

## 4. Discussion

Our results indicates that the *FGFR-3* mutation is among the most common mutated oncogene in urothelial carcinoma, and it is even more common in the LNVUC. In our small study, we report 50% of cases as having a positive activating mutation in *FGFR-3.*

Additionally, we observed that many cases of LNVUC harbor simultaneous multiple activating mutations. In our series, four out of six (66%) LNVUC cases showed multiple activating mutations in oncogenes and truncating mutations in tumor suppressors, simultaneously. Of note, one patient who had two separate primaries of bladder and renal pelvis LNVUC demonstrated different mutational landscape between the two tumors, thus supporting different mutational landscape even in same-patient tumors, based on location.

Based on the targeted *FGFR-3* sequencing of 32 cases of metastatic invasive urothelial carcinoma, 16% of cases showed *FGFR-3* missense mutations. None of the cases that did not harbor *FGFR-3* mutations by whole-exome sequencing showed *FGFR-3* mutations, raising the possibility that point mutations in *FGFR-3* gene are likely more frequent in LNVUC. 

To illustrate our results in relation to published reports using genomic sequencing data, we added a review table for *FGFR-3* analysis and comparison table analysis of two studies, including a study by Pietzak et al. [25] and the TCGA data, to compare the rate of specific mutations across urothelial carcinomas.

As demonstrated in our data, in non-muscle-invasive urothelial carcinoma, our results showed a slight but non-significantly higher incidence of *FGFR-3* mutation, whereas compared to data provided by TCGA, the incident of *FGFR-3* mutations in our variant was significantly higher compared to unselected variants of urothelial carcinoma. The incidence of *CDKN1B*, *GNAS*, *MRE11* and *PPP2R1A* mutations was also significantly higher in our cases compared to both studies. 

Similarly, other studies characterizing non-muscle-invasive and muscle-invasive high-grade urothelial carcinoma reported similar incidence of *FGFR-3* mutations, ranging from 11% to 52% suggesting that the rate of *FGFR-3* mutations may vary significantly depending on the methods used, site of assessment and variants of urothelial carcinoma included in the studies [26,27,28,29] (Table 4).

As in our study, Weyerer et al. [5] focused mainly on the large, nested variant of urothelial carcinoma, but they reported that 97% of their pure nested variants showed *FGFR-3* mutation, whereas only 13% of the mixed tumor variant harbored this mutation. Their finding raises the possibility of different neoplastic pathways for mixed and pure LNVUC in their study.

It is also important to note that LNVUC and advanced UC shows a response to pembrolizumab especially with the recurrent LNVUC; however, our study does not focus on therapeutic strategies but the incidence associated with the presence of FGFR3 mutations [17,18].

Overall, all of these studies document a high incidence of *FGFR-3* mutations in urothelial carcinoma, and they support that the LNVUC variant may exhibit an even higher incidence of *FGFR-3* mutations especially in more pure histological type. Additionally, the rate of *FGFR-3* mutations varies depending on the samples analyzed, whether it is resection or TURBT as well as histological grade, patient’s demographics, and underlying risk factors such as history of smoking. Finally, the studies document that *FGFR-3* mutations usually occur in association with other activating or truncating mutations, especially in the *PI3CKA* pathway, and that LUCNV may harbor simultaneous activating and truncating mutations making them amenable for targeted therapy. 

## 5. Conclusions

Our study provides further evidence of the promising role for *FGFR-3* in the diagnosis and treatment of the large, nested variant of urothelial carcinoma, possibly implicating other targetable pathways compared to random unselected variants of urothelial carcinoma. We do acknowledge the limitations of our study, including small sample size and the fact that most cases were those of TURB. To better evaluate the role of these mutations in this rare variant of urothelial carcinoma, more studies, with a larger number of cases, focused on histomorphology, grade, stage as well as patient demographics and prognosis should be designed. 

## Figures and Tables

**Table 1 cancers-15-03167-t001:** Demographics of the studied cases.

Case	Sex	Age at the Time of Dx	Diagnosis	CIS	Stage at the Time of Dx	Initial Treatment	Progression
1	M	74	Invasive high-grade urothelial carcinoma with features of large, nested variant	No	pT3	Partial cystectomy and 4 cycles of cisplatin and gemcitabine back in January 2016	Progression to stage IV with bone med in 2019
2	M	48	Papillary and inverted urothelial carcinoma with features of large, nested variant of urothelial carcinoma	No	pT2b	Radical cystectomy	Progression with second primary LNVUC
3	M	49	High-grade papillary urothelial carcinoma of kidney	No	pTa	Bilateral nephroureterectomy	Ongoing treatment
4	M	57	High-grade, high volume invasive urothelial carcinoma, with nested areas (predominantly papillary)	Yes	Not done	Bladder preservation Chemotherapy (cisplatin) and radiation, 2018. Pembrolizumab from 2019	No known progression
5	M	61	High-grade urothelial carcinoma, nested variant	No	pT2	Neoadjuvant chemotherapy with cisplatin and gemcitabine November 2018–2019 cystectomy in 2020	No known progression
6	F	73	Invasive urothelial carcinoma, large, nested variant	Yes	pT3a	Adjuvant chemotherapy Jan 2019	No known progression

**Table 2 cancers-15-03167-t002:** Site of tumor primary/metastasis and mutational status of the *FGFR-3* on the additional 32 cases of metastatic invasive urothelial carcinoma for *FGFR-3* using targeted seq.

	Site of Metastasis	Primary Diagnosis	Age	Gender	Mutation
1	Lung	High-grade invasive urothelial carcinoma with focal sarcomatoid differentiation	79	M	Negative
2	Prostate	High-grade invasive urothelial carcinoma with features of large, nested variant infiltrating into muscularis propria and bladder neck	60	M	Negative
3	Penile	High-grade invasive urothelial carcinoma	78	M	Negative
4	Right humerus	Invasive high-grade urothelial carcinoma	72	F	Negative
5	Prostatic urethra	High-grade invasive urothelial carcinoma arising from the prostatic urethra	61	M	FGFR-3 S249C
6	Renal pelvis	Invasive high-grade papillary urothelial carcinoma, squamous differentiation present	71	M	FGFR-3 Y373C
7	Liver	Invasive high-grade papillary urothelial carcinoma	88	M	Negative
8	Lung	Noninvasive high-grade papillary urothelial carcinoma.	71	M	FGFR-3 G370C
9	Lymph node	High-grade invasive urothelial carcinoma	68	M	FGFR-3 Y373C
10	Lymph node	High-grade invasive urothelial carcinoma with extensive squamous differentiation	60	F	Negative
11	Kidney	High-grade urothelial carcinoma (HGUC)	74	M	Negative
12	Lung	Invasive high-grade urothelial carcinoma	66	M	Negative
13	Lymph node, bone, lung and liver	High-grade invasive urothelial carcinoma with focal sarcomatoid differentiation	79	M	Negative
14	Facial bone	Poorly differentiated malignant cells present, compatible with a poorly differentiated carcinoma	49	M	FGFR-3 Y373C
15	Lymph node and liver	High-grade papillary urothelial carcinoma with squamous differentiation	65	M	Negative
16	Lymph node	Urothelial carcinoma, with prominent intraductal spread	77	M	Negative
17	Prostate	High-grade invasive urothelial carcinoma with focal sarcomatoid differentiation,	78	M	FGFR-3 S249C
18	Lymph nodes and peritoneum	High-grade urothelial carcinoma with divergent differentiation	74	M	Negative
19	Bone	High-grade invasive urothelial carcinoma	69	F	Negative
20	Lymph node (para-aortic)	Invasive high-grade urothelial carcinoma	61	M	Negative
21	Lymph node (retroperitoneal)	High-grade invasive urothelial carcinoma arising from the prostatic	66	M	Negative
22	Kidney	Invasive high-grade papillary urothelial carcinoma, sarcomatoid differentiation present	35	F	Negative
23	Liver and bone	Invasive high-grade papillary urothelial carcinoma	66	M	Negative
24	Pelvic soft tissue	Noninvasive high-grade papillary urothelial carcinoma.	64	M	Negative
25	Retroperitoneal soft tissue	High-grade invasive urothelial carcinoma	75	F	Negative
26	Lung	High-grade invasive urothelial carcinoma with extensive squamous differentiation	68	M	Negative
27	Retroperitoneal soft tissue	High-grade urothelial carcinoma (HGUC):	74	M	Negative
28	Retroperitoneal soft tissue	Invasive high-grade urothelial carcinoma	62	M	Negative
29	Lymph node	High-grade invasive urothelial carcinoma with focal sarcomatoid	59	M	Negative
30	None (from the first series)	High-grade, high-volume invasive urothelial carcinoma, with nested areas (predominantly papillary)	57	M	Negative
31	None (from the first series)	High-grade urothelial carcinoma, nested variant	61	M	Negative
32	None (from the first series)	Invasive urothelial carcinoma, large, nested variant	73	F	Negative

**Table 3 cancers-15-03167-t003:** The genetic mutations detected in the 6 cases of LNVUC via WES.

Genes	Frequency	Case 1	Case 2 *	Case 3 *	Case 4	Case 5	Case 6
FGFR-3	50%						
PIK3CA	33%						
CDKN1B	33%						
CDKN2A	17%						
ARID1B	17%						
ARID1A	17%						
GNAS	17%						
MRE11	17%						
KDM6A	17%						
PPP2R1A	17%						
BRD7	17%						
CCDC175	17%						
CFTR	17%						
CNTLN	17%						
CRHR2	17%						
FKBP15	17%						
GPRASP1	17%						
KCNQ3	17%						
KRTAP24-1	17%						
KRTAP24-1	17%						
LOC100129083	17%						
LRP8	17%						
MAGED1	17%						
MBD6	17%						
OR2T2	17%						
OR2T35	17%						
OR6P1	17%						
OR6P1	17%						
PRR30	17%						
PRR30	17%						
RABGGTA	17%						
RBM10	17%						
RREB1	17%						
RYR1	17%						
SIPA1L1	17%						
SMOX	17%						
STX10	17%						
TMC7	17%						
ZNF560	17%						
ZNF560	17%						

* Cases from same patient (case 2, bladder; case 3, renal pelvis). Green Boxes: Activating mutation in oncogenes; Orange Boxes: Truncating mutation in tumor suppressors.

**Table 4 cancers-15-03167-t004:** Data provided on the incidence of *FGFR-3* mutation in our study and reviewed studies.

Study	Method Used	Patient Population	FGFR-3 Mutation
Our study	Whole-genome sequencing	Invasive LNVUC diagnosed on both TURB and cystectomy	50%
	Target sequencing	Metastatic urothelial carcinoma regardless of variant	16%
Pietrzak et al. [25]	Targeted NGS	Non-muscle-invasive UC	49%
The Cancer Genome Atlas (TCGA) 2014 [30]	Whole-exome sequencing	High-grade muscle-invasive urothelial bladder carcinomas	13%
Downes et al. [27]	PCR and SNaPshot methodology	Papillary urothelial carcinoma	52%
Iyer et al. [26]	Review article	Non-muscle-invasive UC	Activating mutation 70%
		Muscle-invasive UC	Overexpression 40%
Al-Ahmadie et al. [28]	Sanger sequencing and MALDI–TOF mass spectrometry	HGUC	17%
		LGUC	84%
Pouessel et al. [29]	PCR-SnaPshot method	T1-TURB UC	38%
		T2-TURB UC	30%
		LN + UC	5%
Weyerer et al. [5]	SnaPshot analysis	Pure LNVUC	94%
		Mixed LNVUC	14.2%
		Overall LNVUC	73.9%

Large, nested urothelial carcinoma (LNVUC), urothelial carcinoma (UC), lymph node (LN), transurethral resection of the bladder (TURB), polymerase chain reaction (PCR).

## Data Availability

The data can be shared up on request.

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
