# Peer review of "Large, Nested Variant of Urothelial Carcinoma Is Enriched with Activating Mutations in Fibroblast Growth Factor Receptor-3 among Other Targetable Mutations"

_cancers, 2023, doi:10.3390/cancers15123167_

Round 1

Reviewer 1 Report (Previous Reviewer 2)

I am afraid that S249C, Y373C, and G370 are neither rearrangement mutations nor fusion mutations but missense mutations.

Please double check the terminology throughout the manuscript.

Author Response

Reviewer 1,

# I am afraid that S249C, Y373C, and G370 are neither rearrangement mutations nor fusion mutations but missense mutations

Answer.Thank you. You are correct. S249C, Y373C, and G370 are missense mutations the typo error was corrected in the revised manuscript.

 Missense mutations are a type of genetic mutation where a single nucleotide change in the DNA sequence results in the substitution of one amino acid for another in the protein encoded by that gene. FGFR3 (Fibroblast Growth Factor Receptor 3) is a gene that encodes a receptor protein involved in cell growth and development. The specific mutations mentioned (S249C, Y373C, and G370) are examples of missense mutations found in the FGFR3 gene. These mutations are specifically changes the amino acid sequence of the FGFR3 protein. Rearrangement mutations involve large-scale rearrangements of genetic material, such as translocations or inversions. Fusion mutations involve the joining of two genes, resulting in a fusion protein. However, the mutations mentioned (S249C, Y373C, and G370) are not indicative of rearrangement or fusion mutations but rather missense mutations.

Reviewer 2 Report (New Reviewer)

The authors presented a case series of FGFR3 mutation in a sample of patients with bladder cancer for large cell nested urothelial variant histology.

It is a rare mixed variant with a poor outcome and unknown specific therapy in case of metastatic spread of disease.

The paper is nice and well written. The role of immune targeted therapy extended to rare variant histology could be a useful treatment option even in this unfavorable disease.

Just a note for authors

In Table 3 is it correct to consider prostatic urethra a site of metastasis in a case starting as urothelial tumor of prostatic urethra? It seems more probable a local disease recurrence. Plase clarify or modify the text.

Further copyediting is needed.

Author Response

Reviewer 2,

The authors presented a case series of FGFR3 mutation in a sample of patients with bladder cancer for large cell nested urothelial variant histology.

It is a rare mixed variant with a poor outcome and unknown specific therapy in case of metastatic spread of disease.

The paper is nice and well written. The role of immune targeted therapy extended to rare variant histology could be a useful treatment option even in this unfavorable disease.

Just a note for authors

# In Table 3 is it correct to consider prostatic urethra a site of metastasis in a case starting as urothelial tumor of prostatic urethra? It seems more probable a local disease recurrence. Please clarify or modify the text.

#Answer We agree with reviewer, this likely represent synchronous tumors as patient had primary bladder carcinoma. We revised the title of the heading to state (site of tumor primary/metastasis).

This manuscript is a resubmission of an earlier submission. The following is a list of the peer review reports and author responses from that submission.

Round 1

Reviewer 1 Report

The presented case series is obviously very small but this histopatological variant could be understimated and less recognized. The mutational scenario of this variant you recognized is however interesting as in this cases a personalized approach. 

Is very interesting the mutational burden of the case presenting with metacronous UTUC in wich two different oncogenesis produced two different tumoral clinical presentation. However we know differences in oncogenesis in the urothelial cancers as bladder cancer and utuc are cousins more than brothers.

Your work is well structured from an oncogenomic point of view.

Some clinical urological data are missing:

_ is there any familiarity, any syndromic presentation?

- is there any work expositional related issue?

- why a partial cistectomy was performed in case 1 and no surgery in case 4

Urological data on the 32 MIBC are lacking ( clinical presentation and treatment)

Probably in this group of metastatic controls the abnormal variants ( sarcomatoid/squamous etc) should be removed considering only mutational scenartio of classical urtothelial MIBC.

Any information about the linfonodal status of these series?

Any evidence in literature about possibility of FRGF-3 mutation in urine citologies?

Pembrolizumab could have a more specific role in this kind of cancers, add some considerations on discussion.

Author Response

Reviewer 1:

The presented case series is obviously very small but this histopathological variant could be underestimated and less recognized. The mutational scenario of this variant you recognized is however interesting as in this cases a personalized approach. 

Is very interesting the mutational burden of the case presenting with metacronous UTUC in wich two different oncogenesis produced two different tumoral clinical presentation. However we know differences in oncogenesis in the urothelial cancers as bladder cancer and utuc are cousins more than brothers.

Your work is well structured from an oncogenomic point of view.

Some clinical urological data are missing:

_ is there any familiarity, any syndromic presentation?

Response: There is no syndromic presentation or any strong family history of such tumors

- is there any work expositional related issue?

Response: As per medical record and clinical histories, no work-related exposure issues

- why a partial cistectomy was performed in case 1 and no surgery in case 4

Response: Due to medical and health concerns, partial cystectomy was performed in case 1 and no surgery performed in case 4 as per clinical managements, but no specific reasons documented.

Urological data on the 32 MIBC are lacking (clinical presentation and treatment)

We thank the reviewer for his comment. The scope of the study is to characterize mutational rate of FGFR in Urothelial carcinoma and its subtypes. We think the therapy and clinical outcome of such cases are beyond the scope of the study and will require significant efforts and will not have any effects on the characterization of FGFR mutation described in this study.

Probably in this group of metastatic controls the abnormal variants ( sarcomatoid/squamous etc) should be removed considering only mutational scenartio of classical urtothelial MIBC.

We thank the reviewer for his comment. However, we think having those subtypes provide added interest into the incidence of FGFR mutations across subtypes or tumor differentiation, so we would prefer to keep this.

Any information about the linfonodal status of these series?

No such data available

Any evidence in literature about possibility of FRGF-3 mutation in urine citologies?

Response: FGFR 3 mutation in voided urine is a useful diagnostic marker and significant indicator of tumor recurrence in non-muscle invasive bladder cancer( pubmed.ncbi.nlm.nih.gov/19843069)

Pembrolizumab could have a more specific role in this kind of cancers, add some considerations on discussion.

 34497992

However, our study didn’t focus on the treatment strategies here and so we didn’t add any regarding.

Reviewer 2 Report

This report shows sequencing data of six tumors of the large, nested variant of urothelial carcinoma, two of which were obtained from one patient at different time points. The number of cases is too small to infer from the results the promising role of FGFR3 in the diagnosis and treatment of the large, nested variant of urothelial carcinoma and to possibly implicate other targetable pathways in the nested variant compared with random unselected variants of urothelial carcinoma.

The authors described that WGS was performed for the six cases of the large, nested variant of urothelial carcinoma in the title of Table 2 and in line 167 of the manuscript, which is inconsistent with the description in the Material and Methods section.

The authors mention that 16% of the cases showed FGFR2/3 rearrangement on targeted sequencing. However, information on FGFR2/3 rearrangement is not present in Table 3 and elsewhere in the manuscript.

Author Response

Reviewer 2

This report shows sequencing data of six tumors of the large, nested variant of urothelial carcinoma, two of which were obtained from one patient at different time points. The number of cases is too small to infer from the results the promising role of FGFR3 in the diagnosis and treatment of the large, nested variant of urothelial carcinoma and to possibly implicate other targetable pathways in the nested variant compared with random unselected variants of urothelial carcinoma.

The authors described that WGS was performed for the six cases of the large, nested variant of urothelial carcinoma in the title of Table 2 and in line 167 of the manuscript, which is inconsistent with the description in the Material and Methods section.

Response: We describes using  WES in our study and not not WGS in the abstract and methods, the description was typo in title of table 2 (this is corrected)

The authors mention that 16% of the cases showed FGFR2/3 rearrangement on targeted sequencing. However, information on FGFR2/3 rearrangement is not present in Table 3 and elsewhere in the manuscript..

The FGFR3 fusion mutation (but no rearrangement status was reported here) available in the last row of table 3 and also described in the results section line 198-199 as follows

“6/32 cases (18%) were positive for FGFR-3 fusion mutations (two S249C; three Y373C and one G370C mutations).

Reviewer 3 Report

1. The abstract is not the introduction. The main content of the full text should be summarized with concise language and accurate words in the abstract section of manuscript.

2. The review is not to list and translate the literature, but to integrate the literature and review the author's own research results. Some of the content of the article is blunt and lacks academic quality.

3. There are too few types of tumors involved in this article, which makes the article unconvincing.

4. References should cite research works, and do not cite review, database analysis, meta-analysis and other types of papers. Moreover, pay attention to the publication time of the article, which should be based on articles in the past three years.

Author Response

Reviewer 3:

  1. The abstract is not the introduction. The main content of the full text should be summarized with concise language and accurate words in the abstract section of manuscript.

Response: The abstract been revised as per reviewer recommendation

  1. The review is not to list and translate the literature, but to integrate the literature and review the author's own research results. Some of the content of the article is blunt and lacks academic quality. Thank you for your suggestion, we have improved this in our revised manuscript.
  2. There are too few types of tumors involved in this article, which makes the article unconvincing.

The LNVUC is uncommon and rare variant but aggressive variant. LNVUC has been added to the World Health Organization (WHO) in 2016. and investigating this was quite hard to accumulate more samples however we involved about 38 samples, 6 was investigated with LNVUC variant and the remaining were UC sample

  1. References should cite research works, and do not cite review, database analysis, meta-analysis and other types of papers. Moreover, pay attention to the publication time of the article, which should be based on articles in the past three years.

All the relative articles been cited. There is only very few articles about LNVUC and especially the FGFR3 in LNVUC